# CPM-Nets: Cross Partial Multi-View Networks

**Changqing Zhang**[1,2]**, Zongbo Han**[1]**, Yajie Cui**[1]**, Huazhu Fu**[3]**, Joey Tianyi Zhou**[4,*]**Qinghua Hu**[1,2]
[1]College of Intelligence and Computing, Tianjin University, Tianjin, China
[2]Tianjin Key Lab of Machine Learning, Tianjin, China
[3]Inception Institute of Artificial Intelligence, Abu Dhabi, UAE
[4]Institute of High Performance Computing, A*STAR, Singapore

## Abstract

Despite multi-view learning progressed fast in past decades, it is still challenging due to the difficulty in modeling complex correlation among different views, especially under the context of view missing. To address the challenge, we propose a novel framework termed Cross Partial Multi-View Networks (CPM-Nets). In this framework, we first give a formal definition of completeness and versatility for multi-view representation and then theoretically prove the versatility of the latent representation learned from our algorithm. To achieve the completeness, the task of learning latent multi-view representation is specifically translated to degradation process through mimicking data transmitting, such that the optimal tradeoff between consistence and complementarity across different views could be achieved. In contrast with methods that either complete missing views or group samples according to view-missing patterns, our model fully exploits all samples and all views to produce structured representation for interpretability. Extensive experimental results validate the effectiveness of our algorithm over existing state-of-the-arts.

## 1   Introduction

In the real-word applications, data is usually represented in different views, including multiple modalities or multiple types of features. A lot of existing methods [1, 2, 3] empirically demonstrate that different views could complete each other, leading ultimate performance improvement. Unfortunately, the unknown and complex correlation among different views often disrupts the integration of different modalities in the model. Moreover, data with missing views further aggravates the modeling difficulty.

Conventional multi-view learning usually holds the assumption that each sample is associated with the unified observed views and all views are available for each sample. However, in practical applications, there are usually incomplete cases for multi-view data [4, 5, 6, 7, 8]. For example, in medical data, different types of examinations are usually conducted for different subjects, and in web analysis, some webs may contain texts, pictures and videos, but others may only contain one or two types, which produce data with missing views. The view-missing patterns (*i.e.*, combinations of available views) become even more complex for the data with more views.

Projecting different views into a common space (*e.g.*, CCA: Canonical Correlation Analysis and its variants [9, 10, 11]) is impeded by view-missing issue. Several methods are proposed to keep on exploiting the correlation of different views. One straightforward way is completing the missing views, and then the on-shelf multi-view learning algorithms could be adopted. The missing views are basically blockwise and thus low-rank based completion [12, 13] is not applicable which has been widely recognized [5, 14]. Missing modality imputation methods [15, 5] usually require samples with two paired modalities to train the networks which can predict the missing modality from the observed one. To explore the complementarity among multiple views, another natural way

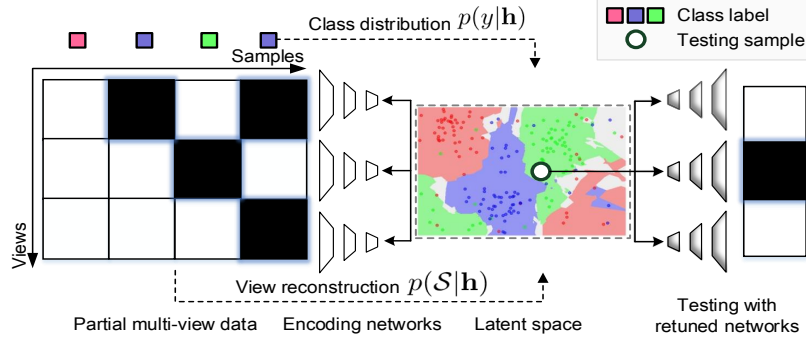

Figure 1: Illustration of Cross Partial Multi-View Networks. Given multi-view data with missing views (black blocks), the encoding networks degrade the complete latent representation into the available views (white blocks). Learning multi-view representation according to the distributions of observations and classes has the promise to encode complementary information, as well as provide accurate prediction.

is manually grouping samples according to the availability of data sources [16], and subsequently learning multiple models on these groups for late fusion. Although it is more effective than learning on each single view, the grouping strategy is not flexible especially for the data with large number of views. Accordingly, a challenging problem arises - how to effectively and flexibly exploit samples with arbitrary view-missing patterns?

Our methodology is expected to endow the following merits: complete and structured representation - comprehensively encoding information from different views into a clustering-structured representation, and flexible integration - handling arbitrary view-missing patterns. To this end, we propose a novel algorithm, *i.e.*, Cross Partial Multi-View Networks (CPM-Nets) for classification, as shown in Fig. 1. Benefiting from the learned common latent representation from the encoding networks, all samples and all views can be jointly exploited regardless of view-missing patterns. For the multi-view representation, CPM-Nets jointly considers multi-view complementarity and class distribution, making them mutually improve each other to obtain the representation reflecting the underlying patterns. Specifically, the encoded latent representation from observations is complete and versatile thus promotes the prediction performance, while the clustering-like classification schema in turn enhances the separability for latent representation. Theoretical analysis and empirical results validate the effectiveness of the proposed CPM-Nets in exploiting partial multi-view data.

## 1.1 Related Work

**Multi-View Learning (MVL)** aims to jointly utilize information from different views. Multi-view clustering algorithms [17, 18, 19, 20, 21] usually search for the consistent clustering hypotheses across different views, where the representative methods include co-regularized based [17], co-training based [18] and high-order multi-view clustering [19]. Under the metric learning framework, multi-view classification methods [22, 23] jointly learn multiple metrics for multiple views. The representative multi-view representation learning methods are CCA based, including kernelized CCA [10], deep neural networks based CCA [11, 24], and semi-paired and semi-supervised generalized correlation analysis ($S^2$GCA) [25]. **Cross-View Learning (CVL)** basically searches mappings between two views, and has been widely applied in real applications [26, 27, 28, 29, 30]. With adversarial training, the embedding spaces of two individual views are learned and aligned simultaneously [27]. The cross-modal convolutional neural networks are regularized to obtain a shared representation which is agnostic of the modality for cross-modal scene images [28]. The cross-view learning can be also utilized for missing view imputation [31, 14]. For **Partial Multi-View Learning (PMVL)**, existing strategies usually transform the incomplete case into complete multi-view learning task. The imputation methods [5, 31] complete the missing views by leveraging the strength of deep neural networks. The grouping strategy [16] divides all samples according to the availability of data sources, and then multiple classifiers are learned for late fusion. Although effective, this strategy cannot scale well for data with large number of views or small-sample-size case. Though the KCCA based algorithm [8] can model incomplete data, it needs one complete (primary) view.

## 2  Cross Partial Multi-View Networks

Recently, there is an increasing interest in learning on data with multiple views, including multi-view learning and cross-view learning. Differently, we focus on classification based on data with missing views, which is termed *Partial Multi-View Classification* (see definition 2.1) where samples with different view-missing patterns are involved. The proposed *cross partial multi-view networks* enable the comparability for samples with different combinations of views instead of samples in two different views, which generalizes the concept of cross-view learning. There are three main challenges for partial multi-view classification: (1) how to project samples with arbitrary view-missing patterns (flexibility) into a common latent space (completeness) for comparability (in section 2.1)? (2) how to make the learned representation to reflect class distribution (structured representation) for separability (in section 2.2)? (3) how to reduce the gap between representation obtained in test stage and training stage for consistency (in section 2.3)? For clarification, we first give the formal definition of partial multi-view classification as follows:

**Definition 2.1** *(**Partial Multi-View Classification (PMVC)**)  Given the training set $\{\mathcal{S}_n, y_n\}_{n=1}^{N}$, where $\mathcal{S}_n$ is a subset of the complete observations $\mathcal{X}_n = \{\mathbf{x}_n^{(v)}\}_{v=1}^{V}$ (i.e., $\mathcal{S} \subseteq \mathcal{X}$) and $y_n$ is the class label with $N$ and $V$ being the number of samples and views, respectively, PMVC trains a classifier by using training data containing view-missing samples, to classify a new instance $\mathcal{S}$ with arbitrary possible view-missing pattern.*

### 2.1  Multi-View Complete Representation

Considering the first challenge - we aim to design a flexible algorithm to project samples with arbitrary view-missing patterns into a common space, where the desired latent representation should encode the information from observed views. Inspired by the reconstruction point of view [32], we provide the definition of completeness for multi-view representation as follows:

**Definition 2.2** *(**Completeness for Multi-View Representation**)  A multi-view representation $\mathbf{h}$ is complete if each observation, i.e., $\mathbf{x}^{(v)}$ from $\{\mathbf{x}^{(1)}, ..., \mathbf{x}^{(V)}\}$, can be reconstructed from a mapping $f_v(\cdot)$, i.e., $\mathbf{x}^{(v)} = f_v(\mathbf{h})$.*

Intuitively, we can reconstruct each view from a complete representation in a numerically stable way. Furthermore, we show that the completeness is achieved under the assumption [33] that each view is conditionally independent given the shared multi-view representation. Similar to each view from $\mathcal{X}$, the class label $y$ can also be considered as one (semantic) view, then we have

$$p(y, \mathcal{S}|\mathbf{h}) = p(y|\mathbf{h})p(\mathcal{S}|\mathbf{h}), \tag{1}$$

where $p(\mathcal{S}|\mathbf{h}) = p(\mathbf{x}^{(1)}|\mathbf{h})p(\mathbf{x}^{(2)}|\mathbf{h})...p(\mathbf{x}^{(V)}|\mathbf{h})$. We can obtain the common representation by maximizing $p(y, \mathcal{S}|\mathbf{h})$.

Based on different views in $\mathcal{S}$, we model the likelihood with respect to $\mathbf{h}$ given observations $\mathcal{S}$ as

$$p(\mathcal{S}|\mathbf{h}) \propto e^{-\Delta(\mathcal{S}, f(\mathbf{h};\mathbf{\Theta}_r))}, \tag{2}$$

where $\mathbf{\Theta}_r$ are parameters governing the reconstruction mapping $f(\cdot)$ from common representation $\mathbf{h}$ to partial observations $\mathcal{S}$ with $\Delta(\mathcal{S}, f(\mathbf{h};\mathbf{\Theta}_r))$ being the reconstruction loss. From the view of class label, we model the likelihood with respect to $\mathbf{h}$ given class label $y$ as

$$p(y|\mathbf{h}) \propto e^{-\Delta(y, g(\mathbf{h};\mathbf{\Theta}_c))}, \tag{3}$$

where $\mathbf{\Theta}_c$ are parameters governing the classification function $g(\cdot)$ based on common representation $\mathbf{h}$, and $\Delta(y, g(\mathbf{h};\mathbf{\Theta}_c))$ defines the classification loss. Accordingly, assuming the data are independent and identically distributed (IID), the log-likelihood function is induced as

$$\mathcal{L}(\{\mathbf{h}_n\}_{n=1}^{N}, \mathbf{\Theta}_r, \mathbf{\Theta}_c) = \sum_{n=1}^{N} \ln p(y_n, \mathcal{S}_n|\mathbf{h}_n) \propto -\Big( \sum_{n=1}^{N} \Delta(\mathcal{S}_n, f(\mathbf{h}_n;\mathbf{\Theta}_r)) + \Delta(y_n, g(\mathbf{h}_n;\mathbf{\Theta}_c)) \Big), \tag{4}$$

where $\mathcal{S}_n$ denotes the available views for the $n$th sample. On one hand, we encode the information from available views into a latent representation $\mathbf{h}_n$ and denote the encoding loss as

$\Delta(\mathcal{S}_n, f(\mathbf{h}_n; \mathbf{\Theta}_r))$. On the other hand, the learned representation should be consistent with class distribution, which is implemented by minimizing the loss $\Delta(y_n, g(\mathbf{h}_n; \mathbf{\Theta}_c))$ to penalize the disagreement with class label.

Effectively encoding information from different views is the key requirement for multi-view representation, thus we seek a common representation which could recover the partial (available) observations. Accordingly, the following loss is induced

$$\Delta(\mathcal{S}_n, f(\mathbf{h}_n; \mathbf{\Theta}_r)) = \ell_r(\mathcal{S}_n, \mathbf{h}_n) = \sum_{v=1}^{V} s_{nv} ||f_v(\mathbf{h}_n; \mathbf{\Theta}_r^{(v)}) - \mathbf{x}_n^{(v)}||^2, \qquad (5)$$

where $\Delta(\mathcal{S}_n, f(\mathbf{h}_n; \mathbf{\Theta}_r))$ is specialized with the reconstruction loss $\ell_r(\mathcal{S}_n, \mathbf{h}_n)$. $s_{nv}$ is an indicator of the availability for the $n$th sample in the $v$th view, $i.e.$, $s_{nv} = 1$ and 0 indicating available and unavailable views, respectively. $f_v(\cdot; \mathbf{\Theta}_r^{(v)})$ is the reconstruction network for the $v$th view parameterized by $\mathbf{\Theta}_r^{(v)}$. In this way, $\mathbf{h}_n$ encodes comprehensive information from different available views, and different samples (regardless of their missing patterns) are associated with representations in a common space, making them comparable.

Ideally, minimizing Eq. (5) will induce a complete representation. Since the complete representation encodes information from different views, it should be versatile compared with each single view. We give the definition of versatility for multi-view representation as follows:

**Definition 2.3** *(Versatility for Multi-View Representation)* *Given the observations* $\mathbf{x}^{(1)}, ..., \mathbf{x}^{(V)}$ *from $V$ views, the multi-view representation $\mathbf{h}$ is of versatility if $\forall$ $v$ and $\forall$ mapping $\varphi(\cdot)$ with $y^{(v)} = \varphi(\mathbf{x}^{(v)})$, there exists a mapping $\psi(\cdot)$ satisfying $y^{(v)} = \psi(\mathbf{h})$, where $\mathbf{h}$ is the corresponding multi-view representation for sample $\mathcal{S} = \{\mathbf{x}^{(1)}, ..., \mathbf{x}^{(V)}\}$.*

Accordingly, we have the following theoretical result:

**Proposition 2.1** *(Versatility for the Multi-View Representation from Eq. (5))* *There exists a solution (with respect to latent representation $\mathbf{h}$) to Eq. (5) which holds the versatility.*

**Proof 2.1** *The proof for proposition 2.1 is as follow. Ideally, according to Eq. (5), there exists $\mathbf{x}^{(v)} = f_v(\mathbf{h}; \mathbf{\Theta}_r^{(v)})$, where $f_v(\cdot)$ is the mapping from $\mathbf{h}$ to $\mathbf{x}^{(v)}$. Hence, $\forall$ $\varphi(\cdot)$ with $y^{(v)} = \varphi(\mathbf{x}^{(v)})$, there exists a mapping $\psi(\cdot)$ satisfying $y^{(v)} = \psi(\mathbf{h})$ by defining $\psi(\cdot) = \varphi(f_v(\cdot))$. This proves the versatility of the latent representation $\mathbf{h}$ based on multi-view observations $\{\mathbf{x}^{(1)}, ..., \mathbf{x}^{(V)}\}$.*

*In practical case, it is usually difficult to guarantee the exact versatility for latent representation, then the goal is to minimize the error $e_y = \sum_{v=1}^{V} ||\psi(\mathbf{h}) - \varphi(\mathbf{x}^{(v)})||^2$ (i.e., $\sum_{v=1}^{V} ||\varphi(f_v(\mathbf{h}; \mathbf{\Theta}^{(v)})) - \varphi(\mathbf{x}^{(v)})||^2$) which is inversely proportional to the degree of versatility. Fortunately, it is easy to show that $Ke_r$ with $e_r = \sum_{v=1}^{V} ||f_v(\mathbf{h}; \mathbf{\Theta}_r^{(v)}) - \mathbf{x}^{(v)}||^2$ from Eq. (5) is the upper bound of $e_y$ if $\varphi(\cdot)$ is Lipschitz continuous with $K$ being the Lipschitz constant.* $\square$

Although the proof is inferred under the condition that all views are available, it is intuitive and easy to generalize the results for view-missing case.

## 2.2 Classification on Structured Latent Representation

Multiclass classification remains challenging due to possible confusing classes [34]. For the second challenge - we target to ensure the learned representation to be structured for separability by a clustering-like loss. Specifically, we should minimize the following classification loss

$$\Delta(y_n, y) = \Delta(y_n, g(\mathbf{h}_n; \mathbf{\Theta}_c)), \qquad (6)$$

where $g(\mathbf{h}_n; \mathbf{\Theta}_c) = \arg\max_{y \in \mathcal{Y}} \mathbb{E}_{\mathbf{h} \sim \mathcal{T}(y)} F(\mathbf{h}, \mathbf{h}_n)$ and $F(\mathbf{h}, \mathbf{h}_n) = \phi(\mathbf{h}; \mathbf{\Theta}_c)^T \phi(\mathbf{h}_n; \mathbf{\Theta}_c)$, with $\phi(\cdot; \mathbf{\Theta}_c)$ being the feature mapping function for $\mathbf{h}$, and $\mathcal{T}(y)$ being the set of latent representation from class $y$. In our implementation, we set $\phi(\mathbf{h}; \mathbf{\Theta}_c) = \mathbf{h}$ for simplicity and effectiveness. By jointly considering classification and representation learning, the misclassification loss is specified as

$$\ell_c(y_n, y, \mathbf{h}_n) = \max_{y \in \mathcal{Y}} \left( 0, \Delta(y_n, y) + \mathbb{E}_{\mathbf{h} \sim \mathcal{T}(y)} F(\mathbf{h}, \mathbf{h}_n) - \mathbb{E}_{\mathbf{h} \sim \mathcal{T}(y_n)} F(\mathbf{h}, \mathbf{h}_n) \right). \qquad (7)$$

---

**Algorithm 1:** Algorithm for CPM-Nets

---
**/\*Training\*/**
**Input**: Partial multi-view dataset: $\mathcal{D} = \{\mathcal{S}_n, y_n\}_{n=1}^N$, hyperparameter $\lambda$.
**Initialize:** Initialize $\{\mathbf{h}_n\}_{n=1}^N$ and $\{\Theta_r^{(v)}\}_{v=1}^V$ with random values.
**while** *not converged* **do**
    **for** $v = 1 : V$ **do**
        Update the network parameters $\Theta_r^{(v)}$ with gradient descent:
        $\Theta_r^{(v)} \leftarrow \Theta_r^{(v)} - \alpha \partial \frac{1}{N} \sum_{n=1}^N \ell_r(\mathcal{S}_n, \mathbf{h}_n; \Theta_r) / \partial \Theta_r^{(v)}$;
    **end**
    **for** $n = 1 : N$ **do**
        Update the latent representation $\mathbf{h}_n$ with gradient descent:
        $\mathbf{h}_n \leftarrow \mathbf{h}_n - \alpha \partial \frac{1}{N} \sum_{n=1}^N (\ell_r(\mathcal{S}_n, \mathbf{h}_n; \Theta_r) + \lambda \ell_c(y_n, y, \mathbf{h}_n)) / \partial \mathbf{h}_n$;
    **end**
**end**

**Output**: networks parameters $\{\Theta_r^{(v)}\}_{v=1}^V$ and latent representation $\{\mathbf{h}_n\}_{n=1}^N$.
**/\*Test\*/**
Train the retuned networks $(\{\Theta_{rt}^{(v)}\}_{v=1}^V)$ for test;
Calculate the latent representation with the retuned networks for test instance;
Classify the test instance with $y = \arg\max_{y \in \mathcal{Y}} \mathbb{E}_{\mathbf{h} \sim \mathcal{T}(y)} F(\mathbf{h}, \mathbf{h}_{test})$.

---

Compared with mostly used parametric classification equipped with cross entropy loss, the clustering-like loss not only penalizes the misclassification but also ensures structured representation. Specifically, for correctly classified sample, *i.e.*, $y = y_n$, there is no loss. For incorrectly classified sample, *i.e.*, $y \neq y_n$, it will enforce the similarity between $\mathbf{h}_n$ and the center corresponding to class $y_n$ larger than that between $\mathbf{h}_n$ and the center corresponding to class $y$ (wrong label) with a margin $\Delta(y_n, y)$. Hence, the proposed nonparametric loss naturally leads to a representation with clustering structure.

Based on above considerations, the overall objective function is induced as

$$\min_{\{\mathbf{h}_n\}_{n=1}^N, \Theta_r} \frac{1}{N} \sum_{n=1}^N \ell_r(\mathcal{S}_n, \mathbf{h}_n; \Theta_r) + \lambda \ell_c(y_n, y, \mathbf{h}_n), \qquad (8)$$

where $\lambda > 0$ balances the belief degree of information from multiple views and class labels.

## 2.3 Test: Towards Consistency with Training Stage

The last challenge lies in the gap between training and test stages in representation learning. To classify a test sample with incomplete views $\mathcal{S}$, we need to obtain its common representation $\mathbf{h}$. A straightforward way is to optimize the objective, $\min_{\mathbf{h}} \ell_r(\mathcal{S}, f(\mathbf{h}; \Theta_r))$, to encode the information from $\mathcal{S}$ into $\mathbf{h}$. This way raises a new issue - how to ensure the representations obtained in test stage consistent with training stage? The gap originates from the difference between the objectives corresponding to training and test stages. Specifically, in test, we can obtain the unified representation with $\mathbf{h} = \arg\min_{\mathbf{h}} \ell_r(\mathcal{S}, f(\mathbf{h}; \Theta_r))$ and then conduct classification with $y = \arg\max_{y \in \mathcal{Y}} \mathbb{E}_{\mathbf{h}_n \sim \mathcal{T}(y)} F(\mathbf{h}, \mathbf{h}_n)$. However, it is different from representation learning in training stage which simultaneously considers reconstruction and classification error. To address this issue, we introduce the fine-tuning strategy based on $\{\mathcal{S}_n, \mathbf{h}_n\}_{n=1}^N$ obtained after training to update the networks $\{f_v(\mathbf{h}; \Theta_r^{(v)})\}_{v=1}^V$ for consistent mapping from observations to latent representation. Accordingly, in test stage we obtain the retuned encoding networks $\{f_v'(\mathbf{h}; \Theta_{rt}^{(v)})\}_{v=1}^V$ by fine-tuning the networks $\{f_v(\mathbf{h}; \Theta_r^{(v)})\}_{v=1}^V$. Subsequently, we can solve the following objective - $\min_{\mathbf{h}} \ell_r(\mathcal{S}, f'(\mathbf{h}; \Theta_{rt}))$ to obtain the latent representation which is consistent with that in training. The optimization of the proposed CPM-Nets and the test procedure are summarized in Algorithm 1.

## 2.4 Discussion on key components

The CPM-Nets are composed of two key components, *i.e.*, encoding networks and clustering-like classification, which are different from conventional ways thus detailed explanations are provided.

**Encoding schema.** To encode the information from multiple views into a common representation, there is an alternative route, *i.e.*, $\ell_r(\mathcal{S}_n, \mathbf{h}_n) = \sum_{v=1}^{V} s_{nv} ||f(\mathbf{x}_n^{(v)}; \mathbf{\Theta}^{(v)}) - \mathbf{h}_n||^2$. This is different from the schema used in our model shown in Eq. (5), *i.e.*, $\ell_r(\mathcal{S}_n, \mathbf{h}_n) = \sum_{v=1}^{V} s_{nv} ||f(\mathbf{h}_n; \mathbf{\Theta}^{(v)}) - \mathbf{x}_n^{(v)}||^2$. The underlying assumption in our model is that information from different views are originated from a latent representation $\mathbf{h}$, and hence it can be mapped to each individual view. Whereas for the alternative, it indicates that the latent representation could be obtained from (mapping) each single view, which is basically not the case in real applications. For the alternative, ideally, minimizing the loss will enforce the representations of different views to be the same, which is not reasonable especially for the views highly independent. From the view of information theory, the encoding network for the $v$th view could be considered as communication channel with fixed property, *i.e.*, $p(\mathbf{x}^{(v)}|\mathbf{h})$ and $p(\mathbf{h}|\mathbf{x}^{(v)})$ for our model and the alternative, respectively, where the degradation process could be mimicked as data transmitting. Therefore, it is more reasonable to send comprehensive information and receive partial information, *i.e.*, $p(\mathbf{x}^{(v)}|\mathbf{h})$ compared with its counterpart - sending partial data and receiving comprehensive data, *i.e.*, $p(\mathbf{h}|\mathbf{x}^{(v)})$. The theoretical results in subsection 2.1 also advocates above analysis.

**Classification model.** For classification, the widely used strategy is to learn a classification function based on $\mathbf{h}$, *i.e.*, $y = f(\mathbf{h}; \mathbf{\Theta})$ parameterized with $\mathbf{\Theta}$. Compared with this manner, the reasons of using the clustering-like classifier in our model are as follows. First, jointly learning the latent representation and parameterized classifier is likely an under-constrained problem which may find representation that can well fit the training data but not well reflect the underlying patterns, thus the generalization ability may be affected [35]. Second, the clustering-like classification produces the compactness within the same class and separability between different classes for the learned representation, making the classifier interpretable. Third, the nonparametric way reduces the load of parameter tuning and reflects a simpler inductive bias which is especially beneficial to small-sample-size regime [36].

## 3 Experiments

### 3.1 Experiment Setting

We conduct experiments on the following datasets: ⋄ **ORL** [2] The dataset contains 10 facial images for each of 40 subjects. ⋄ **PIE** [3] A subset containing 680 facial images of 68 subjects are used. ⋄ **YaleB** Similar to previous work [37], we use a subset which contains 650 images of 10 subjects. For ORL, PIE and YaleB, three types of features: intensity, LBP and Gabor are extracted. ⋄ **CUB** [38] The dataset contains different categories of birds, where the first 10 categories are used and deep visual features from GoogLeNet and text features using doc2vec [39] are used as two views. ⋄ **Handwritten** [4] The dataset contains 10 categories from digits '0' to '9', and 200 images in each category with 6 types of image features are used. ⋄ **Animal** The dataset consists of 10158 images from 50 classes with two types of deep features extracted with DECAF [40] and VGG19 [41].

We compared the proposed CPM-Nets with the following methods: (1) **FeatConcate** simply concatenates multiple types of features from different views. (2) **CCA** [9] maps multiple types of features into one common space, and subsequently concatenates the low-dimensional features of different views. (3) **DCCA** (Deep Canonical Correlation Analysis) [11] learns low-dimensional features with neural networks and concatenates them. (4) **DCCAE** (Deep Canonical Correlated AutoEncoders) [24] employs autoencoders for common representations, and then combines these projected low-dimensional features together. (5) **KCCA** (Kernelized CCA) [10] employs feature mappings induced by positive-definite kernels. (6) **MDcR** (Multi-view Dimensionality co-Reduction) [42] applies the kernel matching to regularize the dependence across multiple views and projects each view onto a low-dimensional space. (7) **DMF-MVC** (Deep Semi-NMF for Multi-View Clustering) [43] utilizes a deep structure through semi-nonnegative matrix factorization to seek a common feature representation. (8) **ITML** (Information-Theoretic Metric Learning) [44] characterizes the metric using a Mahalanobis distance function and solves the problem as a particular Bregman optimization. (9) **LMNN** (Large Margin Nearest Neighbors) [45] searches a Mahalanobis distance metric to optimize the $k$-nearest neighbours classifier. For metric learning methods, the original features

of multiple views are concatenated, and then the new representation could be obtained with the projection induced by the learned metric matrix.

For all methods, we tune the parameters with 5-fold cross validation. For CCA-based methods, we select two views for the best performance. For our CPM-Nets, we set the dimensionality ($K$) of the latent representation from $\{64, 128, 256\}$ and tune the parameter $\lambda$ from the set $\{0.1, 1, 10\}$ for all datasets. We run 10 times for each method to report the mean values and standard deviations. Please refer to the supplementary material for the details of network architectures and parameter settings.

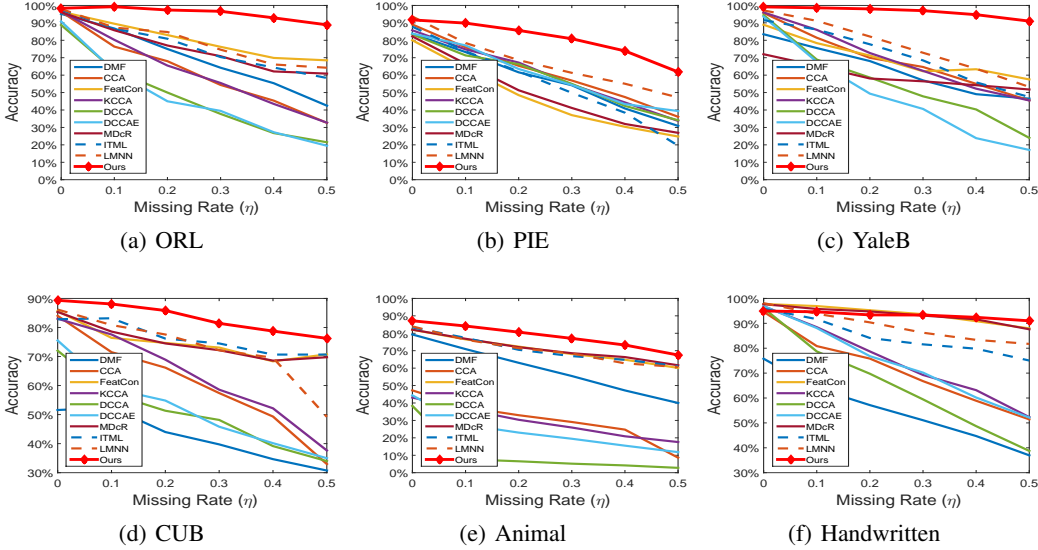

Figure 2: Performance comparison under different missing rate ($\eta$).

## 3.2 Experimental Results

Firstly, we evaluate our algorithm by comparing it with state-of-the-art multi-view representation learning methods, investigating the performance with respect to varying missing rate. The missing rate is defined as $\eta = \frac{\sum_v M_v}{V \times N}$, where $M_v$ indicates the number of samples without the $v$th view. Since datasets may be associated with different number of views, samples are randomly selected as missing multi-view ones, and the missing views are randomly selected by guaranteeing at least one of them is available. As a result, partial multi-view data are obtained with diverse missing patterns. For compared methods, the missing views are filled with mean values according to available samples within the same class. From the results in Fig. 2, we have the following observations: (1) without missing, our algorithm achieves very competitive performance on all datasets which validates the stability of our algorithm for complete multi-view data; (2) with increasing the missing rate, the performance degradations of the compared methods are much larger than that of ours. Taking the results on ORL for example, ours and LMNN obtain the accuracy of 98.4% and 98.0%, respectively, while with increasing the missing rate, the performance gap becomes much larger; (3) our model is rather robust to view-missing data, since our algorithm usually performs relatively promising with heavily missing cases. For example, the performance decline (on ORL) is less than 5% with increasing the missing rate from $\eta = 0.0$ to $\eta = 0.3$.

Furthermore, we also fill the missing views with recently proposed imputation method - Cascaded Residual Autoencoder (CRA) [5]. Since CRA needs a subset of samples with complete views in training, we set 50% data as complete-view samples and the left are samples with missing views (missing rate $\eta = 0.5$). The comparison results are shown in Fig. 3. It is observed that filling with CRA is generally better than that of using mean values due to capturing the correlation of different views. Although the missing views are filled with CRA by using part of samples with complete views, our proposed algorithm still demonstrates the clear superiority. The proposed CPM-Nets performs as the best on all the six datasets.

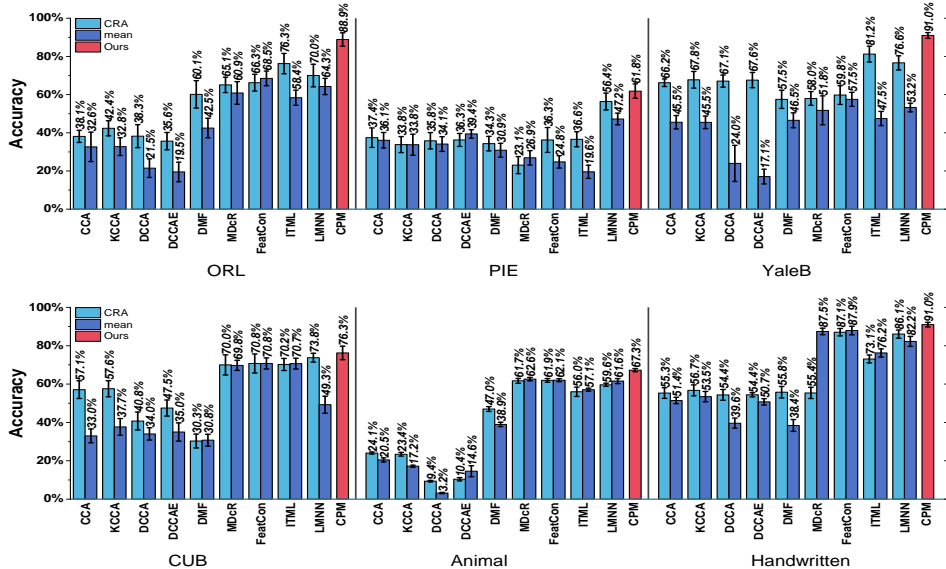

Figure 3: Performance comparison with view completion by using mean value and cascaded residual autoencoder (CRA) [5] (with missing rate $\eta = 0.5$).

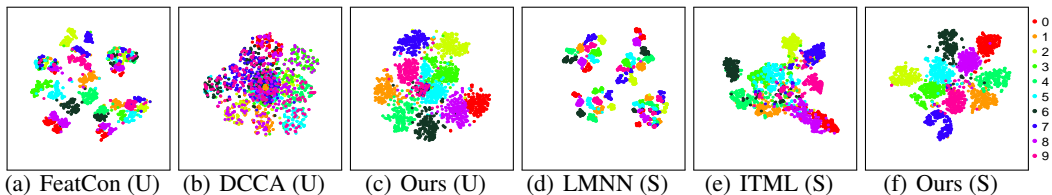

| (a) FeatCon (U) | (b) DCCA (U) | (c) Ours (U) | (d) LMNN (S) | (e) ITML (S) | (f) Ours (S) |

Figure 4: Visualization of representations with missing rate $\eta = 0.5$, where 'U' and 'S' indicate 'unsupervised' and 'supervised' manner in representation learning. (Zoom in for best view).

We visualize the representations from different methods on Handwritten to investigate the improvement of CPM-Nets. As shown in Fig. 4, the subfigures (a)-(c) obtain representations in unsupervised manner. It is observed that the latent representation from our algorithm reveals the underlying class distribution much better. With introducing label information, the representation from CPM-Nets are further improved, where the clusters are more compact and the margins between different classes becomes more clear, which validates the effectiveness of using clustering-like loss. It is noteworthy that we jointly exploit all samples, all views for random view-missing patterns in experiments, demonstrating the flexility in handling partial multi-view data, while Fig. 4 supports the claim of structured representation.

## 4 Conclusions

We proposed a novel algorithm for partial multi-view data classification named CPM-Nets, which can jointly exploit all samples, all views and is flexible for arbitrary view-missing patterns. Our algorithm focuses on learning a complete thus versatile representation to handling the complex correlation among multiple views. The common representation also endows the flexibility for handling the data with arbitrary number of views and complex view-missing patterns, which is different from existing ad hoc methods. Equipped with a clustering-like classification loss, the learned representation is well structured making the classifier interpretable. We empirically validated that the proposed algorithm is relatively robust to heavy and complex view-missing data.

## Acknowledgments

This work was partly supported by National Natural Science Foundation of China (61976151, 61602337, 61732011, 61702358). We also appreciate the discussion with Ganbin Zhou and valuable comments from all the reviewers.

## Footnotes

*Corresponding author: J. T. Zhou <joey.tianyi.zhou@gmail.com>.

[2]https://www.cl.cam.ac.uk/research/dtg/attarchive/facedatabase.html

[3]http://www.cs.cmu.edu/afs/cs/project/PIE/MultiPie/Multi-Pie/Home.html

[4]https://archive.ics.uci.edu/ml/datasets/Multiple+Features

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
