[Supplementary Material]

# Supplementary Material for CPM-Nets: Cross Partial Multi-View Networks

## 1 Network architectures and parameter setting

For our CPM-Nets, we employ the fully connected networks equipped with sigmoid activation for all datasets, and $\ell_2$-norm regularization is used with the value of tradeoff parameter being $0.001$. The dimensionalities of the input, hidden and output layers are denoted as $K$, $M$ and $D$, respectively. The dimensionality of the input layer (*i.e.*, the dimensionality of the latent representation) is selected from the set $\{64, 128, 256\}$. The main architectures are the same with detailed difference described as follows:

- **ORL**. We employ 3-layer (*i.e.*, input/hidden/ouput layers) fully connected networks. The dimensionalities of the input, hidden and output layers are $K = 256$, $M = 300$ and $D$ (4096, 3304 and 6750 respectively for three views). The learning rate is set as $0.01$.

- **PIE**. We employ 3-layer (*i.e.*, input/hidden/ouput layers) fully connected networks. The dimensionalities of the input, hidden and output layers are $K = 256$, $M = 300$ and $D$ (484, 256 and 279 respectively for three views). The learning rate is set as $0.01$.

- **YaleB**. We employ 3-layer (*i.e.*, input/hidden/ouput layers) fully connected networks. The dimensionalities of the input, hidden and output layers are $K = 128$, $M = 350$ and $D$ (2500, 3304 and 6750 respectively for three views). The learning rate is set as $0.01$.

- **Handwritten**. We employ 3-layer (*i.e.*, input/hidden/ouput layers) fully connected networks. The dimensionalities of the input, hidden and output layers are $K = 64$, $M = 200$ and $D$ (240, 76, 216, 47, 64 and 6 respectively for six views). The learning rate is set as $0.001$.

- **CUB**. We employ 2-layer (*i.e.*, input/ouput layers) fully connected networks. The dimensionalities of the input and output layers are $K = 128$, and $D$ (1024 and 300 respectively for two views). The learning rate is set as $0.01$.

- **Animal**. We employ 4-layer (*i.e.*, input/2 hidden/ouput layers) fully connected networks. The dimensionalities of the input, hidden and output layers are $K = 256$, $M$ (512 and 1024 respectively for two hidden layers) and $D$ (4096 for two views). The learning rate is set as $0.001$.

We note that promising performance could be expected with relatively shallow networks for most datasets, *i.e.*, the number of network layers is set as $2 \sim 4$ for all datasets.

## 2 Results with varying the number of views

In practice, the number of views may be different in different applications, which motivates us to investigate the effect of varying the number of views. Fig. 1 gives the classification results with different number of views on multiple datasets, where the views are generated by dividing the original features into multiple nonoverlapping subsets. Specifically, each view is divided into $T$ (from $\{2,4,6,8\}$) views. The experiments are conducted with missing rate $\eta = 0.5$. Generally, the performance from the proposed CPM-Nets is improved with increasing the number of views. The possible reason is that the distribution of missing features for each sample tend to be uniformly across the original views by dividing one view into more views, which alleviates the block-wise (view-wise) missing issue to some extend.

Figure 1: Performance comparison with respect to different number of views, where each original view is divided into $T$ views.

## 3 Parameter tuning and convergence

There are two main parameters in our algorithm, *i.e.*, the tradeoff factor ($\lambda$) and the dimensionality ($K$) of the latent representation. We visualize the parameter tuning on Handwritten (with missing rate $\eta = 0.5$) as shown in Fig. 2. We tune the parameters from fine-grained sets, *i.e.*, selecting $\lambda$ from $\{0, 0.1, ..., 0.9, 1, 2, ..., 10\}$ and the dimensionality ($K$) from the set $\{16, 32, 64, 128, 256\}$. 5-fold cross-validation is employed with the mean values of accuracy reported.

Figure 2: Parameter tuning.

For relatively small datasets, the model parameters of CPM-Nets could be updated by processing the whole training data. For large datasets, it could be solved in the way of mini-batch training. As shown in Fig. 3, we empirically investigate the convergence property on the dataset Animal with mini-batch gradient descent. For each mini-batch, 10% samples from each class are used.

Figure 3: Convergence curve on dataset.