[Reviews · NeurIPS 2019]

Reviewer 1



The authors picked a relevant problem and the approach they propose looks reasonable. I have two major concerns:(i) the methodology assumes that the latent representation encodes similarities between networks. This can behave very different from what the authors intended (see Donnat and Holmes in AOAS) since different measures of similarity on graph space may lead to radically different results when clustering; (ii) There is literature for modelling multiple networks via latent variables. It is not clear how the proposed method relates to established approaches (Gollini and Murphy, Nielsen and Witten). The results from the real data sets look promising.

Reviewer 2



This paper proposes a novel algorithm to utilize the data with missing views, and the algorithm shows the superior performance. Strength: 1) The proposed strategy of handling view-missing is novel and elegant compared with existing strategies. I agree that most existing methods [5][8][16] are not flexible especially for the data with large number of views and complex missing patterns. 2) The theoretical analysis of completeness and versatility is a good support for the success of the learned multi-view representation, which is inspiring for the field of multi-view learning. 3) The designed clustering-like loss is interesting and a good choice for small-sample-size data. Moreover, the conducted experiments are extensive, and the proposed model achieves better results consistently. Minor comments: (1) It will be better if the authors could provide analysis about the computational cost, and release their code, since there may be potential applications for the method. (2) In Figure 4, different clusters correspond to different digits, so can you label the clusters with corresponding digits?

Reviewer 3



The paper proposed a multi-view learning framework that is able to handle missing views. The improvement compared to state-of-the-art seem to be the preservation of versatility with the new formulation, and better numerical results shown in section 3. I increased my rating after reading the authors' feedback. The paper is clear and related algorithmic steps/analysis are well presented. I like the discussion on the consistence and complementarity while establishing the framework. The technical part (1)-(5) is intuitive and easy to follow, although the existence prove in prop 2.1 does not bear much guarentee on the versatility of (5), which requires a more in-depth study of its solution property. Together with the results in supplement, the evaluation part is comprehensive with alternatives compared and discussed. I would encourage the authors to share their implementation for reproducibility and future works on partial multi-view learning.

[Author Response · NeurIPS 2019]



**+Q1**. The methodology assumes that the latent representation encodes similarities between networks. This can behave very different from what the authors intended (see Donnat and Holmes in AOAS) since different measures of similarity on graph space may lead to radically different results when clustering.

**Reply**: Thanks for the comment. The proposed methodology breaks the assumption that different views (associated with different neural networks) should be highly similar or correlated. Instead, our model aims to encode both consistency (similarities) and complementarity (dissimilarities) information from different views, that is why we term the learned representation as complete latent representation. The discussion about encoding schema (lines 181-196) also supports the claim.

As for the work (Donnat and Holmes) in AOAS, graphs with different measures (*e.g.*, Hamming and Jaccard distances) yield very different results capturing different characteristics, hence, these graphs may be little similar or correlated. For our method in handling these graphs, the following aspects should be clarified: (1) Since the proposed model can well balance between consistency (similarities) and complementarity (dissimilarities) across different views, it is suitable to deal with these graphs regardless whether they are similar. (2) To use the proposed model, the graphs could be transformed into feature vectors. It is possible to directly handle both graphs and feature vectors within a unified framework (which is inspired by the next comment).

**+Q2**: Statistical approaches (Gollini and Murphy, Nielsen and Witten) modeling multiple networks.

**Reply:** Thanks for the suggestion. The relationships between the proposed method and the work mentioned by the reviewer are: (1) **Different tasks**: The methods (Gollini and Murphy, Nielsen and Witten) aim to model the networks (*i.e.*, graphs) with latent variables, while our model focuses on the partial multi-view classification task. (2) **Different assumptions**: For the work (Gollini and Murphy, Nielsen and Witten), the underlying assumption is that the smaller the distance between two nodes in the latent space, the greater their probability of being connected. Differently, the proposed model is based on the point of view of reconstruction, encoding the intrinsic information from multiple views for the complete representation and versatility. (3) **Possible connection**: Both the methods (Gollini and Murphyalso, Nielsen and Witten) and our method model multiple sources with latent variables. It is very interesting to propose a more general framework which can handle highly heterogeneous data, *e.g.*, vector-valued and network views.

**+Q1**: Theoretical/empirical results about the computational cost.

**Reply**: Thanks for the suggestion. The computational complexity of our algorithm is basically $O(kn^2 + cn^2)$, where $n$, $k$ and $c$ is the number of samples, the dimensionality of the latent representation and the number of classes, respectively. We have conducted experiments on the Animal dataset and the computational times are reported in Table 1. All these methods are tested on a computer with 4 GPUs (TITAN Xp). It is observed that the efficiency of the proposed method is competitive with existing methods. Limited by space, detailed analysis for computational complexity will be added into the supplement.

Table 1: Computational cost (in seconds).

| Methods | Ours | DCCA | DCCAE | MDcR | DMF |
|---------|------|------|-------|------|-----|
| Time | 270.8 | 390.7 | 518.5 | 628.6 | 189.0 |

**+Q2**: Label the clusters with corresponding digits.

**Reply**: Thanks for the suggestion. We will label each cluster in Fig. 4 to improve the visualization.

**+Q3**: Release the code, since there may be potential applications for the method.

**Reply**: The code is ready and will be released after acceptance. All the results reported are reproducible.

**+Q1**: More in-depth study of the solution property.

**Reply**: Thanks for the suggestion. We will clarify this in the revised version: (1) We provided the analysis for both ideal and practical cases, *i.e.*, perfect reconstruction and under reconstruction. Specifically, we provided strict proof and upper bound of the versatility for ideal and practical cases, respectively. (2) Although the proof is inferred under the condition that all views are available, it is intuitive and easy to generalize the results (*i.e.*, completeness and versatility) for view-missing case in Eq. (5). This is mainly because that different views are decoupled in both the definition and proof of versatility, then the proof still holds for the view-missing case.

**+Q2**: Share the implementation for reproducibility and future works on partial multi-view learning.

**Reply**: The code is ready and will be released after acceptance. All the results reported are reproducible.

[Meta-Review · NeurIPS 2019]

The reviewers all agree that the proposed method is novel, the writing clear and the experiments sufficient. The theoretical analysis of completeness and versatility is a good support for the empirical success. The author's response was very clear and addressed most of the reviewers' concerns.